# Assessment of Fruit Quality and Genes Related to Proanthocyanidins Biosynthesis and Stress Resistance in Persimmon (*Diospyros kaki* Thunb.)

Sichao Yang [1], Meng Zhang [2], Ming Zeng [1], Meihua Wu [1], Qinglin Zhang [2,*], Zhengrong Luo [2]  and Xinlong Hu [1,*]

1   Horticultural Research Institute, Jiangxi Academy of Agricultural Sciences, Nanchang 330200, China
2   Key Laboratory of Horticultural Plant Biology, College of Horticulture and Forestry Sciences, Huazhong Agricultural University, Wuhan 430070, China
*   Correspondence: zhangqinglin@mail.hzau.edu.cn (Q.Z.); h15870008656@126.com (X.H.)

**Abstract:** Persimmon (*Diospyros kaki* Thunb.) is becoming a fruit that is used worldwide because it contains high nutritional and medicinal value. However, the trait evaluation of persimmon is still needed and is critical for marketing and breeding, especially in China. Here, we evaluated thirteen quality indicators (fruit weight, horizontal length, vertical length, firmness, titratable acid content, vitamin C content, flavonoid content, anthocyanin content, soluble sugar content, pectinase activity, soluble protein content, tannin content, and tannin cell size) in six Chinese pollination-constant, astringent (PCA) persimmon cultivars ('Jinxi 3′, 'Ganfang 1′, 'Poyang 5′, 'Poyang 6′, 'Yifeng 1′, and 'Yifeng 3′) and a Japanese pollination-constant non-astringent (PCNA) persimmon cultivar ('Youhou'). The quality indicator data were normalized and subjected to variant analysis and principal component analysis (PCA). The results showed that 10 of 13 indicators among the seven persimmon cultivars were significantly different, and the contribution rates of the first principal component reached 40.582%. The principal component comprehensive scores for 'Poyang 5′ and 'Ganfang 1′ ranked second and third, respectively, and were clustered with that of 'Youhou.' In addition, we also measured the expression levels of three stress resistance genes and three proanthocyanidin (PA, also called condensed tannins) pathway genes in these persimmon cultivars by quantitative reverse transcription PCR (qRT—PCR). The qRT—PCR analysis of *DkCBF* and *DkWRKY3/8* showed low resistance to cold in 'Ganfang 1′ but stronger resistance to anthracnose. Moreover, the expression of the PA pathway genes demonstrated that the PA content in 'Ganfang 1′ was at a moderate level in the seven varieties. Together, our study revealed relatively comprehensive profiles of persimmon quality evaluation and demonstrated that 'Ganfang 1′ may have the potential to be used as a breeding parent for future persimmon breeding programs.

**Keywords:** persimmon; *Diospyros kaki*; variant analysis; principal component analysis; comprehensive scores; proanthocyanidins; stress resistance; quality assessment



## 1. Introduction

Persimmon (*Diospyros kaki* Thunb.), which originated in southern China and was disseminated to Korea and Japan centuries ago, belongs to the genus *Diospyros* in the family Ebenaceae [1]. Now, it is also widely cultivated in Brazil, Spain, Turkey, Italy, Israel, and New Zealand. In addition to *D. kaki*, this genus also includes *D. lotus*, *D. glaucifolia*, *D. rhombifolia*, *D. cathayensis*, *D. oleifera*, and *D. virginiana*, which are usually used as rootstocks in most cases. In China, persimmon is one of the most important fruit trees and is also called "woody food" and a "hardcore crop" [2,3]. FAO statistics have reported that the production and cultivation area of persimmon in China ranked first in the world in 2020 at 3,247,068 t (approximately 76.040% of the global production) and 919,995 ha (approximately 92.700% of the global cultivation area), respectively [FAOSTAT, 2021, www.fao.org/faostat/en/ (accessed on 4 August 2022)].

At the current stage, the breeding goals for persimmon emphasize improved fruit appearance quality, such as fruit weight, fruit shape, skin color, and fruit cracking; enhanced fruit interior quality, such as fruit texture, soluble solids content (SSC), fruit flavor, and fruit astringence quality; prolonged fruit shelf and storage life; ameliorated fruit ripening time; increased productivity; selected parthenocarpy and female-flower-only sexual reproduction patterns; and the expanded commercial use of fruit (consumption as fresh or dried fruit or use in ornamental and industrial applications) [2]. Moreover, there are several obstacles to persimmon breeding and improvement in China. First, almost all cultivated persimmons are the astringent type. Second, it is urgently necessary to select suitable breeding parents for persimmon breeding improvement. Third, the biotechnologies and candidate genes needed for further genetic improvement are lacking.

Fruit quality is an important influencing factor for variety selection and consumption and includes considerations of exterior and interior quality [4]. The former includes fruit weight, fruit size, fruit shape index, and fruit firmness, and the latter is the major component that includes the content of titratable acid, vitamin C, soluble sugar, anthocyanins, flavonoids, soluble protein, tannins, and so on [5]. In addition, pectin is a major component of the cytoderm, which is related to fruit softening and the loss of astringency in persimmon fruit and is another important index of fruit quality [6]. Currently, the methods for evaluating comprehensive traits include principal component analysis and cluster analysis [5,7]. In addition, persimmon fruit contains abundant tannins, and its astringency is mainly caused by the synthesis of a type of tannins known as PAs. The PA-specific pathway is the last pathway in PA biosynthesis and includes *DkANR* (anthocyanidin reductase), *DkLAR* (leucoanthocyanidin reductase), and *DkLAC* (laccase) [8]. In this study, we chose three genes involved in PA biosynthesis to compare the tannin accumulation mechanisms in seven persimmon cultivars.

Jiangxi Province is one of the traditional persimmon production areas in southern China, and it has rich persimmon germplasms and native varieties [9–11]. The persimmon resources in Jiangxi can be divided into four classes according to fruit shape: square persimmon, round persimmon, flat persimmon, and heart-shaped persimmon. These resources include eight species and varieties according to plant classification: *D. kaki*, *D. oleifera*, *D. silvestris* Makino, *D. nitida* Merr., *D. lotus*, *D. glaucifolia* Metc., *Diospyros morrisiana* Hance, and Yongding red persimmon. According to investigations of the germplasms in Jiangxi over the last four years, there are 8 varieties and 37 native varieties (Supplementary Materials Table S1) [10]. Through traditional cross-breeding processes, outstanding cultivars that can adapt to Jiangxi farming conditions are urgently needed. Our study comprehensively evaluates the quality of six persimmon varieties in Jiangxi Province and one Japanese variety, PCNA (J-PCNA), at the fruit ripening stage and analyzes the expression level of the genes involved in PA biosynthesis mechanisms and responses to adversity and stress. The results of the present study may help with expanding the range of breeding parents for the PCNA persimmon, which would have great potential for the genetic improvement of persimmon cultivars adapted to cultivation in southern China.

## 2. Materials and Methods

### 2.1. Materials

Persimmon fruits were sampled from the persimmon repository at the Horticultural Institute, Jiangxi Academy of Agricultural Sciences, Nanchang, China, in November 2020; the climate and soil conditions from where the samples derived are shown in the Supplementary Materials Figure S1 and Table S2, respectively. Fruits without visible defects that had uniform size were collected from three trees of each variety with a random orientation; every sixth fruit was a repeat, and each variety was repeated three times. The fruits were frozen in liquid nitrogen and stored at -80 °C after measuring the exterior quality. The varieties *D. kaki* 'Jinxi 3′, *D. kaki* 'Ganfang 1′, *D. kaki* 'Poyang 5′, *D. kaki* 'Poyang 6′, *D. kaki* 'Yifeng 1′, and *D. kaki* 'Yifeng 3′ are native astringent varieties in Jiangxi; *D. kaki* 'Youhou' is a J-PCNA persimmon, and the tree age was 4 years old (Figure 1). The budding period is around 24 February, and the flowering period is around April 26 every year. The trees

are in open trellis and the growth potential is strong with only female flowers, except for 'Yifeng 1' and 'Yifeng 3', which are monoecious and both bear only female flowers and male flowers. In this study, all the following tests were conducted with fresh fruit, and weight is expressed by fresh weight (FW).

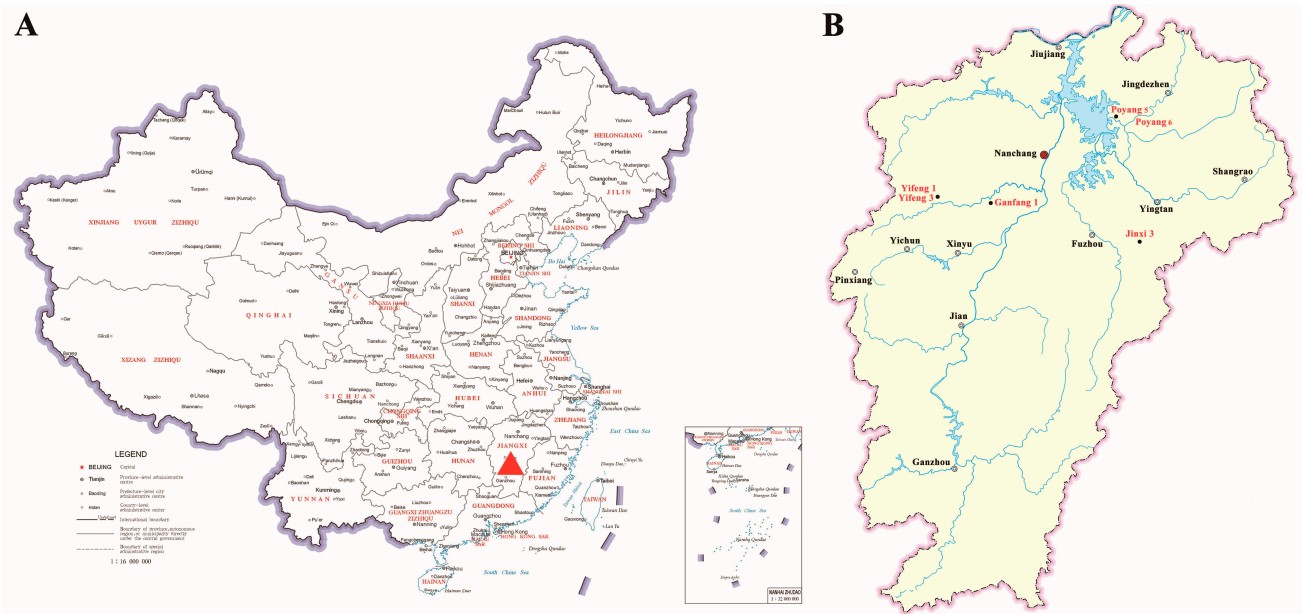

**Figure 1.** Distribution of 'Ganfang 1' and five other *D. kaki* cultivar samples from different regions in Jiangxi Province. (**A**) Location of Jiangxi Province on the map of China. (**B**) Distribution of different persimmon germplasms (in red color) in Jiangxi Province.

### 2.2. Measurement of the Exterior Quality of Fruit

The fruit weight and horizontal/vertical lengths were measured with an electronic balance and Vernier calipers, respectively. The fruit firmness was measured with a hardness tester, and each fruit measurement was repeated three times.

### 2.3. Measurement of the Interior Quality of Fruit

The fruit flesh was ground into serous fluid and then the supernatant of the filtrate was absorbed to measure the major components. The titratable acid content was measured by acid-base titration according to the methods of [12,13]. The extracting solution was titrated with 0.1 mol/L NaOH standard solution after phenolphthalein reagent was added.

The vitamin C content was determined by 2,6-dichloroindophenol titration according to [14], and the vitamin C content was calculated based on the ascorbic acid equivalent.

The soluble sugar content was measured by anthrone colorimetry according to the methods of [15]. The extract was added with 1 mL 9% phenol solution and 5 mL concentrated sulfuric acid successively. After standing at room temperature for 30 min, the color was measured at 485 nm.

The soluble protein content was calculated by the Coomassie brilliant blue method [16]. Take 1 mL extract, add 5 mL Coomassie Brilliant Blue G-250, mix and stand for 2 min, and measure absorbance at 595 nm.

The anthocyanin and flavonoid levels were measured by UV spectrophotometer colorimetry [17]. The flesh was extracted with 1% hydrochloric acid-methanol solution and the absorbance was measured at 325 nm and 530 nm. The contents of flavonoids and anthocyanins were calculated, respectively.

The soluble and insoluble tannin levels were detected by the Folin–Ciocalteu method according to [18].

The pectinase activity was quantified by UV spectrophotometer colorimetry [19]. The pulp was grinded into homogenate and mixed with 2 mL of 0.4% pectin solution. The

mixture was reacted in a 45 °C water bath for 30 min. Then, 1.5 mL of 3,5-Dinitrosalicylic acid (DNS) reagent was added and heated in a boiling water bath for 5 min. After cooling, the solution was diluted to 25 mL. The absorbance was measured at 520 nm. The amount of D-galacturonic acid was determined based on the standard curve.

The tannin cell area was measured according to [20]. The observation of the paraffin section of fruits was performed according to the methods of [21].

### 2.4. Statistical Analysis of Data

The correlation analysis, principal component analysis, and cluster analysis were performed with SPSS 25.0 software (IBM, Stanford, CA, USA). The chart was created with Origin 9.1 (OriginLab, Northampton, MA, USA) and Adobe Photoshop CC 2021 (Adobe, San Jose, CA, USA).

### 2.5. RNA Isolation and cDNA Synthesis

The total RNA was isolated from the fruit flesh using RNAiso Plus* (Tiangen, Beijing, China). The RNA quality and quantity were assessed by gel electrophoresis and Nanodrop 2000 spectrophotometry (Thermo Scientific, Madison, WI, USA). Three biological replicates were performed for each sample. For gene isolation, first-strand cDNA was generated using a 2.0-µg total RNA sample with the PrimeScript TM RT reagent kit with gDNA Eraser (Takara, Dalian, China) according to the manufacturer's protocol. For gene expression analysis, cDNA was synthesized from 1.0 µg of each RNA sample using the PrimeScript RT Kit with gDNA eraser (TaKaRa, Dalian, China).

### 2.6. qRT—PCR

Quantitative reverse transcription PCR (qRT—PCR) was performed with a QuantStudio 7 Flex Real-Time PCR system (Applied Biosystems, Thermo Fisher Scientific, Singapore) using SYBR® Premix Ex TaqTM II (TaKaRa, Dalian, China). The PCR mixture (10 µL total volume) included 5 µL of SYBR Premix Ex Taq II (TaKaRa, Dalian, China), 3.5 µL of $H_2O$, 1.0 µL of diluted cDNA, and 0.25 µL of each primer (0.01 M). The PCR conditions were as follows: 5 min at 95 °C, 45 cycles of 95 °C for 5 s, 58 °C for 10 s, and 72 °C for 15 s. Each sample was assayed in quadruplicate, and *DkActin* (accession no. AB473616) was used as the internal control (Supplementary Materials Table S3). All treatments were performed with at least three biological replicates.

## 3. Results and Discussion

### 3.1. Analysis of Variation and the Quality of Different Persimmon Fruit

The variation coefficient of the quality indices of the seven persimmon types was substantial; the soluble sugar content had the highest variation coefficient (71.190), followed by the tannin content (57.180). Furthermore, the vitamin C, pectinase activity, soluble protein, tannin cell size, flavonoid content, fruit weight, and firmness variation coefficients were all greater than 20%, indicating a substantial difference in the seven indices in the seven persimmons (Table 1).

As shown in Figure 2, there was a large difference in fruit weight. Among the seven varieties, the heaviest cultivar, 'Yifeng 1', was 186.712 g, followed by 'Ganfang 1' at 177.350 g, and the smallest cultivars were 'Jinxi 3' at 136.308 g and 'Poyang 6' at 52.446 g (Figure 2A,B). The horizontal fruit length was the longest in 'Youhou' at 76.941 mm, followed by 76.732 mm in 'Ganfang 1'. The horizontal fruit length was the shortest in 'Poyang 6' at 44.676 mm, followed by 'Jinxi 3' at 68.893 mm (Figure 2C). The vertical fruit length was the longest in 'Yifeng 1' at 60.770 mm and the shortest, at 50.932 mm, in 'Poyang 6'. The firmness of the fruit was greatest in 'Youhou', and that of 'Jinxi 3' was the lowest (Figure 2D). The shape of fruits, such as apples, pears, and peaches, is known to be affected by the seasonality of the vertical fruit length and horizontal fruit length. The fruit shape index of the seven persimmon varieties ranged from 0.687 to 1.140, and with the exception of 'Poyang 6', all varieties had a fruit shape index less than 1 (Figure 2E).

**Table 1.** Correlation analysis among major quality parameters of the different persimmon cultivars.

| Genotypes | Exterior Quality | | | | Interior Quality | | | | | | | | |
|---|---|---|---|---|---|---|---|---|---|---|---|---|---|
| | Fruit Weight (g) | Horizontal Length (mm) | Vertical Length (mm) | Firmness (kg/cm²) | Titratable Acidity (FW%) | Vitamin C (mg/100 g FW) | Soluble Sugar (FW%) | Pectinase Activity (U/g FW) | Anthocyanin Content (mg/g FW) | Flavonoid Content (mg/g FW) | Soluble Protein (mg/g FW) | Tannin Cell Size (μm²) | Tannin Content (FW%) |
| Jinxi 3 | 136.308 | 68.893 | 53.191 | 1.833 | 1.831 | 268.094 | 0.005 | 0.053 | 0.004 | 2.375 | 0.497 | 15,243.906 | 6.882 |
| Ganfang 1 | 177.350 | 76.732 | 57.919 | 2.433 | 1.973 | 209.524 | 0.038 | 0.090 | 0.007 | 4.543 | 0.496 | 9914.305 | 11.768 |
| Poyang 5 | 165.439 | 71.554 | 60.700 | 2.567 | 2.052 | 238.893 | 0.051 | 0.068 | 0.008 | 4.979 | 0.980 | 7391.723 | 5.217 |
| Poyang 6 | 52.446 | 44.676 | 50.932 | 2.250 | 2.047 | 100.477 | 0.036 | 0.018 | 0.007 | 4.038 | 0.423 | 15,047.179 | 14.550 |
| Yifeng 1 | 186.712 | 75.431 | 60.770 | 3.056 | 1.845 | 192.857 | 0.122 | 0.084 | 0.005 | 2.102 | 0.523 | 16,809.189 | 7.353 |
| Yifeng 3 | 141.181 | 69.650 | 55.230 | 2.144 | 1.867 | 166.667 | 0.131 | 0.050 | 0.003 | 1.611 | 0.535 | 14,216.744 | 14.685 |
| Youhou | 159.523 | 76.941 | 52.788 | 3.500 | 1.771 | 260.476 | 0.116 | 0.039 | 0.002 | 1.387 | 0.986 | 3995.812 | 1.235 |
| Average value | 145.570 | 69.130 | 55.930 | 2.540 | 1.910 | 205.280 | 0.071 | 0.057 | 0.005 | 3.005 | 0.630 | 11,802.690 | 8.810 |
| Standard deviation | 44.850 | 11.270 | 3.940 | 0.569 | 0.110 | 58.840 | 0.051 | 0.025 | 0.002 | 1.478 | 0.240 | 4782.460 | 5.040 |
| Coefficient of variation (%) | 30.810 | 16.310 | 7.040 | 22.380 | 5.830 | 28.660 | 71.190 | 43.860 | 40.000 | 49.185 | 37.990 | 40.520 | 57.180 |

Note: All indicators were measured using fresh fruit.

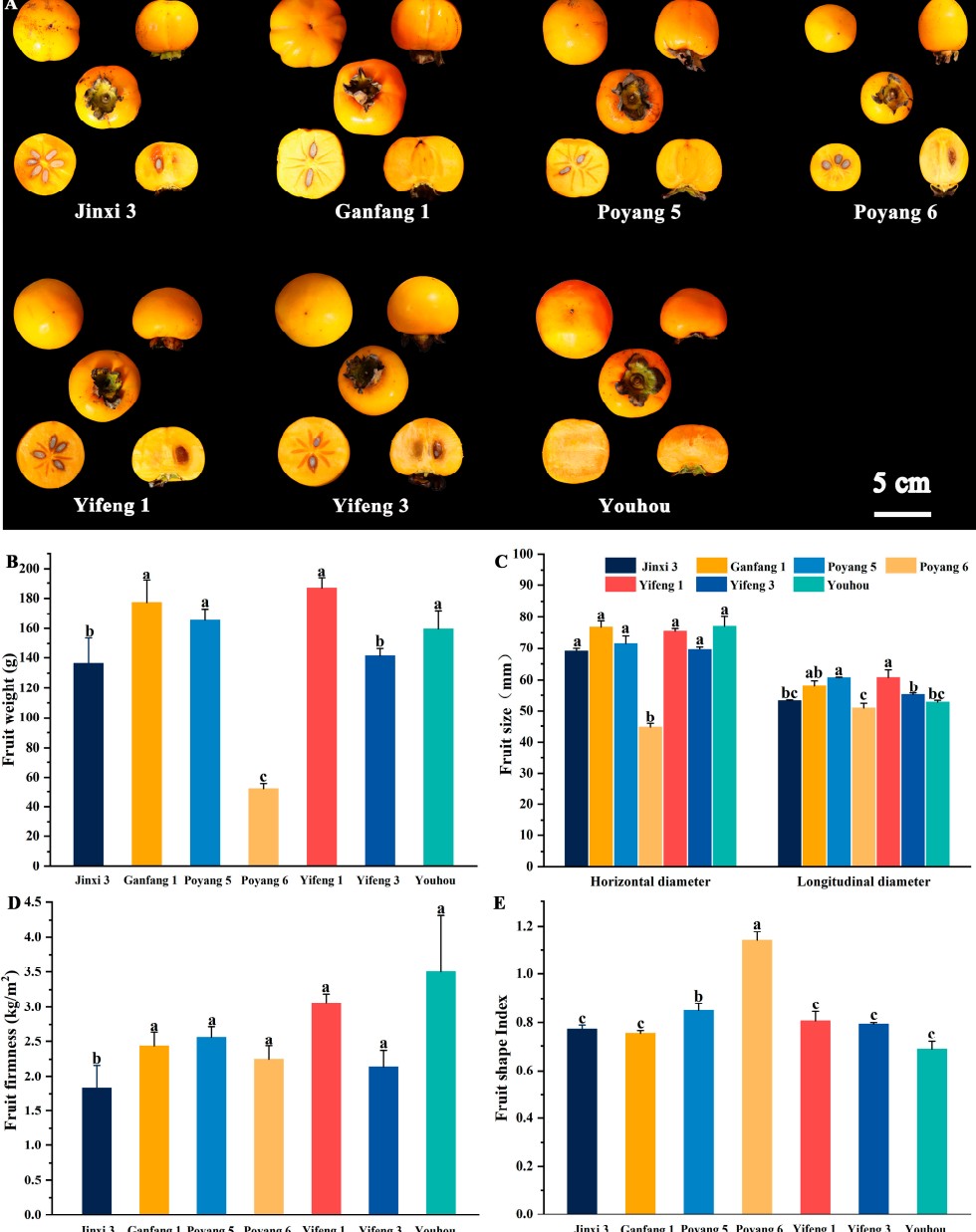

**Figure 2.** Measurement of the exterior quality of different persimmon cultivars. (**A**) Photograph of persimmon fruits of different persimmon genotypes at 25 WAB (weeks after bloom). (**B**) Fruit weight of different persimmon genotypes at 25 WAB. (**C**) Fruit size of different persimmon genotypes at 25 WAB. (**D**) Fruit firmness of different persimmon genotypes at 25 WAB. (**E**) Fruit shape index of different persimmon genotypes at 25 WAB. All indicators were measured using fresh fruit. Means with different letters indicate significant differences at $p < 0.05$ by Tukey's test.

As shown in Figure 3, the titratable acid content of the seven persimmon varieties ranged from 1.771 FW% to 2.052 FW%; that of 'Poyang 5′ was the highest, and that of 'Youhou' was the lowest (Figure 3A). The vitamin C content of the seven persimmon varieties ranged from 100.477 to 268.094 mg/100 g FW; that of 'Jinxi 3′ was the highest, and that of 'Poyang 6′ was the lowest (Figure 3B). The soluble sugar content of the seven persimmon varieties ranged from 0.005 FW% to 0.131 FW%; that of 'Yifeng 3′ was the highest, and that of 'Jinxi 3′ was the lowest (Figure 3C). The soluble protein content of the seven persimmon varieties ranged from 0.423 to 0.986 mg/g FW; that of 'Youhou' was the highest, and that of 'Poyang 6′ was the lowest (Figure 3D). The anthocyanidin content of the seven persimmon

varieties ranged from 0.002 to 0.008 mg/g FW; that of 'Poyang 5' was the highest, and that of 'Youhou' was the lowest (Figure 3E). The flavonoid content of the seven persimmon varieties ranged from 1.387 to 4.979 mg/g FW; that of 'Poyang 5' was the highest, and that of 'Youhou' was the lowest (Figure 3F). The higher the pectinase activity was, the lower the pectin content, and the fruit softened easily. As shown in Figure 4, the pectinase activity of the fruit of the seven persimmon varieties was significantly different, ranging from 0.018 to 0.090 U/g FW; that of 'Ganfang 1' was the highest, and that of 'Poyang 6' was the lowest.

The tannin cell area in the fruit of the seven persimmon varieties ranged from 3995.812 to 16,809.189 μm$^2$; that of 'Yifeng 1' was the largest, and that of 'Youhou' was the smallest (Figure 5A1–G1). The soluble tannin content ranged from 0.485 to 9.489 mg/g FW; that of 'Poyang 6' was the highest, and that of 'Youhou' was the lowest (Figure 6A). The insoluble tannin content was between 0.750 mg/g FW and 6.172 mg/g FW; that of 'Ganfang 1' was the highest, and that of 'Youhou' was the lowest (Figure 6B). The paraffin section of flesh stained with toluidine blue showed the same results as the tannin content; the tannin cell color of 'Youhou' was the lightest, and that of 'Ganfang 1' was the darkest (Figure 5A2–G2).

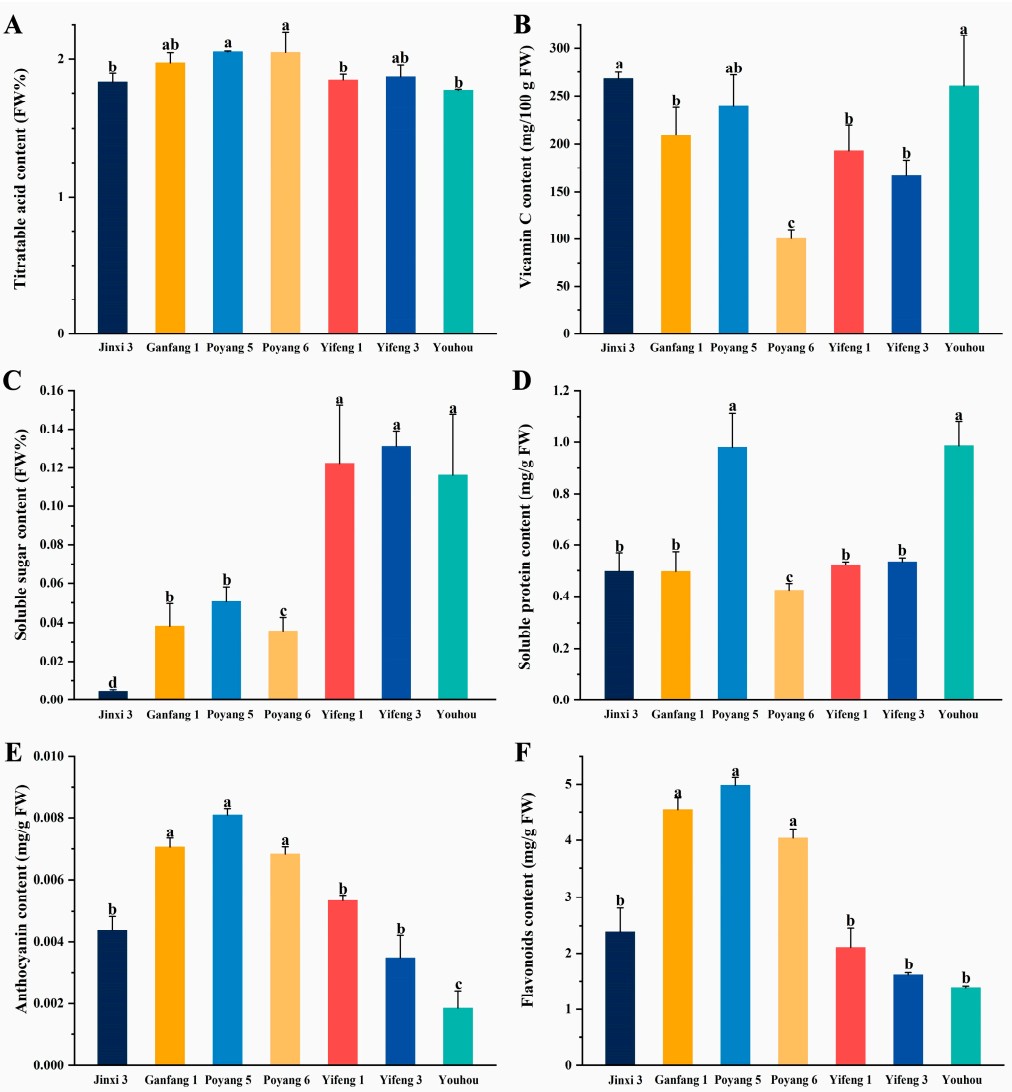

**Figure 3.** Measurement of the interior quality in fruits of different persimmon cultivars. (**A**) Titratable acid content. (**B**) Vitamin C content. (**C**) Soluble sugar content. (**D**) Soluble protein content. (**E**) Anthocyanin content. (**F**) Flavonoid content. Means with different letters indicate significant differences at *p* < 0.05 by Tukey's test. All indicators were measured using fresh fruit.

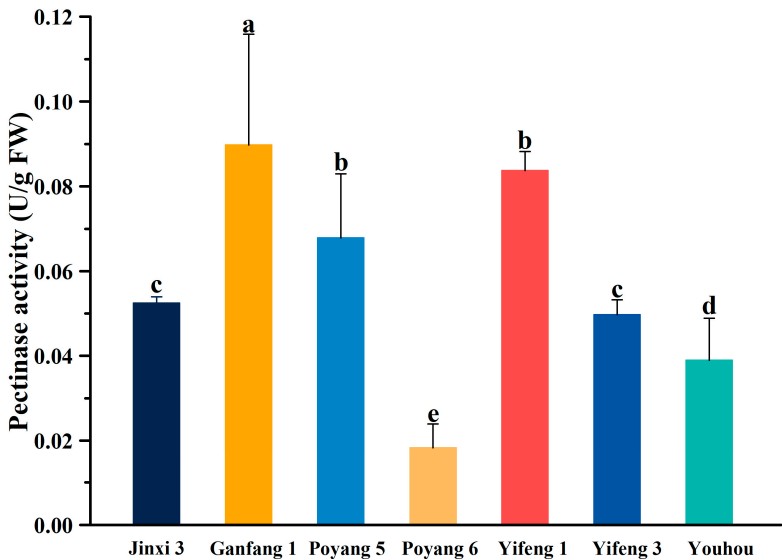

**Figure 4.** Measurement of pectinase activity in fruits of different persimmon cultivars. Means with different letters indicate significant differences at *p* < 0.05 by Tukey's test. All indicators were measured using fresh fruit.

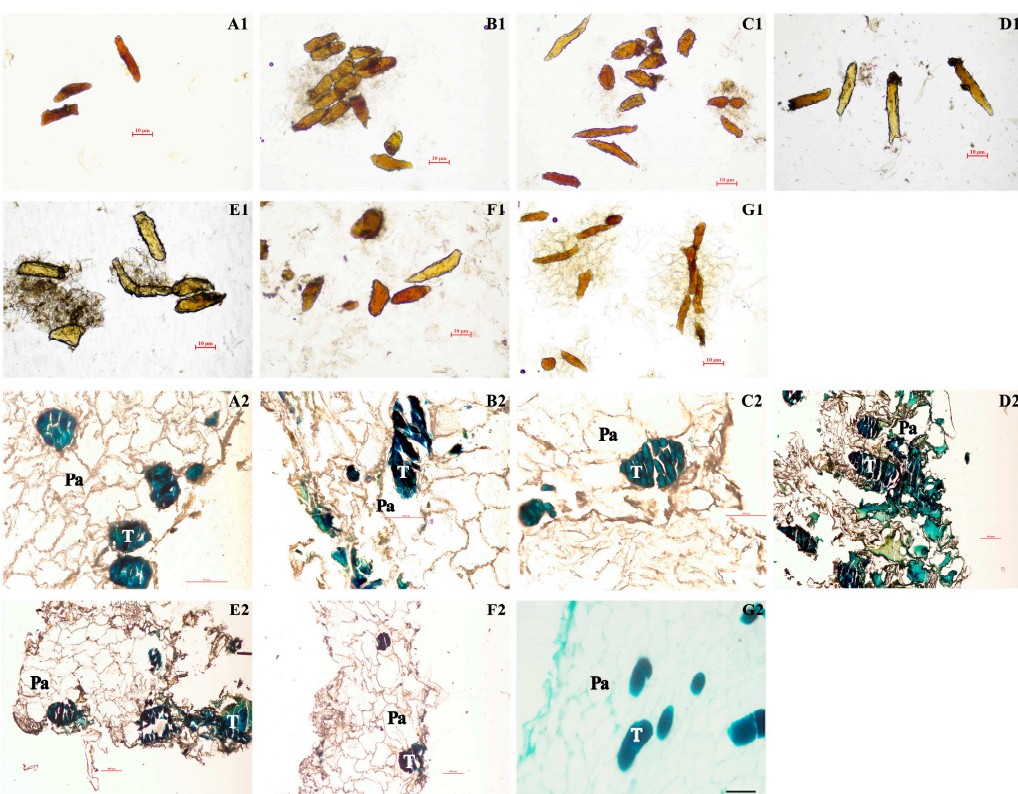

**Figure 5.** The morphology of tannin cells in fruits of different persimmon cultivars. (**A1**,**A2**) 'Jinxi 3'; (**B1**,**B2**) 'Ganfang 1'; (**C1**,**C2**) 'Poyang 5'; (**D1**,**D2**) 'Poyang 6'; (**E1**,**E2**) 'Yifeng 1'; (**F1**,**F2**) 'Yifeng 3'; (**G1**,**G2**) 'Youhou.' (**A1**–**G1**) observation of the tannin cells in fruits of different types of persimmons at the ripe stage under an upright microscope (OLYMPUS BX61); (**A2**–**G2**) paraffin section observation of the tannin cells stained with toluidine blue. T = Tannin Cell, Pa = Parenchyma. All indicators were measured using fresh fruit.

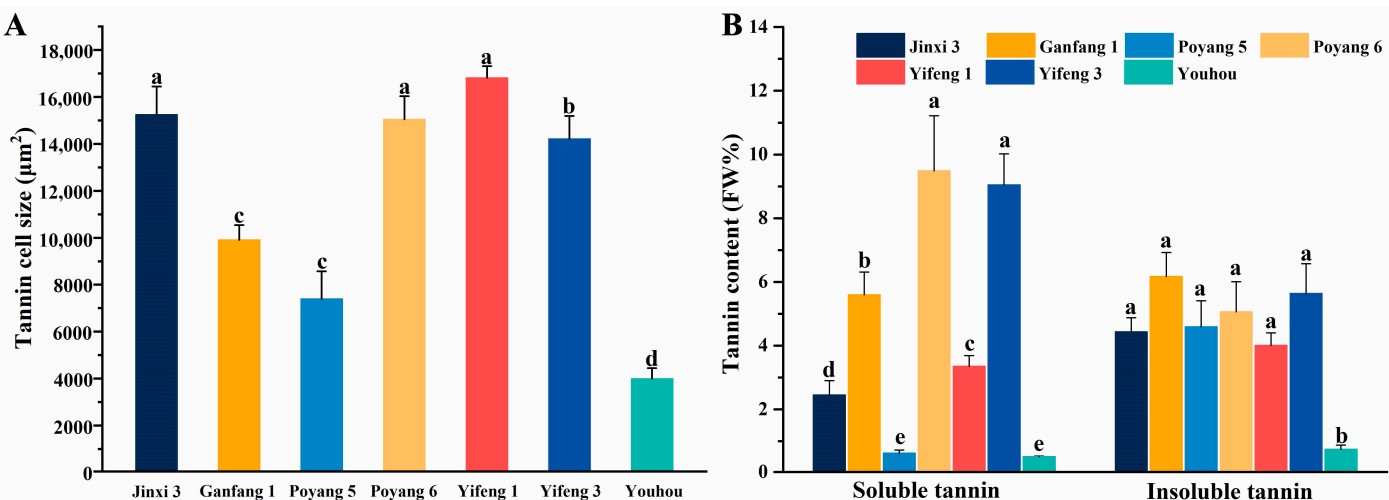

**Figure 6.** Measurement of the tannin cell size (**A**) and tannin content (**B**) in fruits of different persimmon cultivars. Means with different letters indicate significant differences at $p < 0.05$ by Tukey's test. All indicators were measured using fresh fruit.

### 3.2. Correlation Analysis of Indicators

Principal component analysis (PCA) can simplify the number of evaluation indicators while preserving most indicator information. Multiple original indicators were converted into a reduced number of independent indicators, making it easier to establish a quality evaluation method for persimmons [22]. Zx1 to Zx13 represent the normalized data for each of the 13 indicators (fruit weight, horizontal length, vertical length, firmness, titratable acid content, vitamin C content, flavonoid content, anthocyanin content, soluble sugar content, pectinase activity, soluble protein content, tannin content, and tannin cell size). The correlations between the normalized indicators were calculated before PCA in SPSS 25.0 and are listed in Table 2. These results indicated that there was a positive correlation between the horizontal fruit length and fruit weight, and between titratable acidity and flavonoid content ($p < 0.01$). There was a significant positive correlation between fruit weight and vertical fruit length or pectinase activity, between fruit vertical length and pectinase activity, between titratable acidity and anthocyanin content, and between flavonoid content and anthocyanin content ($p < 0.05$). There was a significant negative correlation between firmness and soluble sugar content, between vitamin C content and tannin content, and between soluble protein content and tannin content or tannin cell size ($p < 0.05$).

**Table 2.** Correlation analysis among major quality parameters of different persimmon cultivars.

| | | Tx1 | Tx2 | Tx3 | Tx4 | Tx5 | Tx6 | Tx7 | Tx8 | Tx9 | Tx10 | Tx11 | Tx12 | Tx13 |
|---|---|---|---|---|---|---|---|---|---|---|---|---|---|---|
| | | | | | | Correlation Matrix | | | | | | | | |
| **Trait** | | **Tx1** | **Tx2** | **Tx3** | **Tx4** | **Tx5** | **Tx6** | **Tx7** | **Tx8** | **Tx9** | **Tx10** | **Tx11** | **Tx12** | **Tx13** |
| Fruit weight (g FW) | Tx1 | 1 | | | | | | | | | | | | |
| Horizontal length (mm) | Tx2 | 0.967 ** | 1 | | | | | | | | | | | |
| Vertical length (mm) | Tx3 | 0.762 * | 0.583 | 1 | | | | | | | | | | |
| Firmness (kg/cm$^2$) | Tx4 | −0.500 | −0.443 | −0.568 | 1 | | | | | | | | | |
| Titratable acid content (FW%) | Tx5 | −0.388 | −0.534 | 0.185 | 0.069 | 1 | | | | | | | | |
| Vitamin C content (mg/100 g FW) | Tx6 | 0.657 | 0.752 | 0.226 | 0.114 | −0.495 | 1 | | | | | | | |
| Flavonoid content (mg/g FW) | Tx7 | −0.144 | −0.285 | 0.428 | 0.18 | 0.928 ** | −0.206 | 1 | | | | | | |
| Anthocyanin content (mg/g FW) | Tx8 | −0.444 | −0.532 | 0.033 | 0.439 | 0.786 * | −0.476 | 0.818 * | 1 | | | | | |
| Soluble sugar content (FW%) | Tx9 | 0.361 | 0.375 | 0.217 | −0.835 * | −0.494 | −0.990 | −0.571 | −0.052 | 1 | | | | |
| Pectinase activity (U/g FW) | Tx10 | 0.855 * | 0.734 | 0.863 * | −0.348 | −0.020 | 0.368 | 0.219 | 0.269 | 0.061 | 1 | | | |
| Soluble protein content (mg/g FW) | Tx11 | 0.372 | 0.427 | 0.216 | −0.326 | −0.093 | 0.578 | −0.575 | 0.248 | 0.248 | −0.085 | 1 | | |
| Tannin content (FW%) | Tx12 | −0.506 | −0.562 | −0.185 | −0.031 | 0.448 | −0.820 * | 0.639 | −0.105 | −0.105 | −0.035 | −0.780 * | 1 | |
| Tannin cell size (μm$^2$) | Tx13 | −0.307 | −0.412 | −0.030 | 0.205 | 0.026 | −0.513 | 0.459 | −0.097 | −0.097 | −0.042 | −0.863 * | 0.611 | 1 |

Note: * means at *p* = 0. 05 level significantly correlated, ** means at *p* = 0. 01 level significantly correlated.

### 3.3. Principal Component Analysis of Fruit Quality

The correlation analysis indicated that there was a degree of information overlap among the 13 indicators, which highlighted the utility of a PCA approach. In Table 3, four principal components were retained, with eigenvalues of e1 = 5.276, e2 = 3.316, e3 = 2.337, and e4 = 1.578. The cumulative variance of the four principal components was 96.209%, which included most of the information from the 13 original indicators. In the first principal component (PC1), the contribution rate was 40.582%, and the variance contributions of fruit horizontal length, fruit weight, vitamin C content, tannin content, soluble protein content, and titratable acid content occupied dominant positions with larger absolute values, indicating that this principal component was influenced mainly by fruit horizontal length, fruit weight, vitamin C content, tannin content, soluble protein content, and titratable acid content. The contribution rate of principal component 2 (PC2) was 25.506%, which was strongly influenced by the anthocyanin content, flavonoid content, fruit vertical length, titratable acid content, and pectinase activity. The contribution rate of principal component 3 (PC3) was 17.980%, which was influenced by tannin cell size and fruit firmness. The contribution rate of principal component 4 was 12.141%, which was strongly influenced by fruit firmness and soluble protein content.

**Table 3.** Eigenvectors and contribution rates of the four principal components.

| Index | Eigenvectors | | | |
|---|---|---|---|---|
| | Principal Component 1 | Principal Component 2 | Principal Component 3 | Principal Component 4 |
| Fruit weight | 0.889 | 0.351 | 0.204 | 0.202 |
| Horizontal length | 0.939 | 0.173 | 0.095 | 0.205 |
| Vertical length | 0.491 | 0.743 | 0.382 | 0.017 |
| Firmness | −0.502 | −0.149 | −0.598 | 0.596 |
| Titratable acid content | −0.623 | 0.709 | −0.089 | −0.309 |
| Vitamin C content | 0.759 | 0.046 | −0.505 | 0.367 |
| Soluble sugar | 0.533 | −0.362 | 0.567 | −0.497 |
| Pectinase activity | 0.523 | 0.653 | 0.370 | 0.385 |
| Anthocyanin content | −0.468 | 0.869 | −0.075 | −0.060 |
| Flavonoid content | −0.424 | 0.859 | −0.256 | −0.093 |
| Soluble protein content | 0.629 | 0.115 | −0.538 | −0.524 |
| Tannin cell size | −0.513 | −0.140 | 0.609 | 0.477 |
| Tannin content | −0.728 | 0.038 | 0.575 | −0.025 |
| Eigenvalues | 5.276 | 3.316 | 2.337 | 1.578 |
| Contribution rates (%) | 40.582 | 25.506 | 17.980 | 12.141 |
| Cumulative contribution rates (%) | 40.582 | 66.088 | 84.068 | 96.209 |

### 3.4. Comprehensive Evaluation of Different Persimmon Fruits

We can establish a comprehensive evaluation model for fruit quality based on the variance contribution rates of each principal component (Table 4).

$$Z = 0.4058Z1 + 0.2551Z2 + 0.1798Z3 + 0.1214Z4$$

Z1, Z2, Z3, and Z4 were the scores for the principal components and were calculated using the equation and Table 4. From the total comprehensive scores in Table 4, it can be observed that 'Youhou' > 'Poyang 5′ > 'Ganfang 1′ > 'Yifeng 3′ > 'Jinxi 3′ > 'Poyang 6′ > 'Yifeng 1′, indicating that the fruit quality of 'Poyang 5′ and 'Ganfang 1′ ranked second and third among the seven varieties.

**Table 4.** Principal component score and a comprehensive evaluation of different persimmon cultivars.

| Cultivar | Principal Component Comprehensive Score | | | | Total Score | Ranking |
|---|---|---|---|---|---|---|
| | Z1 | Z2 | Z3 | Z4 | | |
| Jinxi 3 | 7413.229 | −2019.359 | 9203.670 | 7413.180 | −968.647 | 5 |
| Ganfang 1 | −4681.491 | −1254.332 | 6001.277 | 4858.517 | −550.876 | 3 |
| Poyang 5 | −3374.207 | −902.564 | 4444.376 | 3662.245 | −355.802 | 2 |
| Poyang 6 | −7544.180 | −2032.932 | 9152.512 | 7235.214 | −1056.053 | 6 |
| Yifeng 1 | −8218.115 | −2217.529 | 10,209.935 | 8143.000 | −1076.296 | 7 |
| Yifeng 3 | −6961.686 | −1876.996 | 8636.920 | 6885.962 | −915.000 | 4 |
| Youhou | −1615.278 | −436.677 | 2360.218 | 2050.675 | −93.557 | 1 |

### 3.5. Cluster Analysis of Different Persimmon Fruits

We comprehensively analyzed the exterior and interior quality of the different persimmon fruits utilizing principal component analysis. To explore the similarity of principal components in the different varieties, normalized data from 13 indicators measured in each of the seven persimmon cultivars were used for cluster analysis with SPSS software and a between-group linkage clustering method. The results showed that the persimmon varieties could be divided into the following two classes: the first class contained 'Jinxi 3', 'Poyang 6', 'Yifeng 3', and 'Yifeng 1'; and the second class included 'Ganfang 1', 'Poyang 5', and 'Youhou', implying that the major components of 'Ganfang 1' and 'Poyang 5' were similar to 'Youhou' (Figure 7).

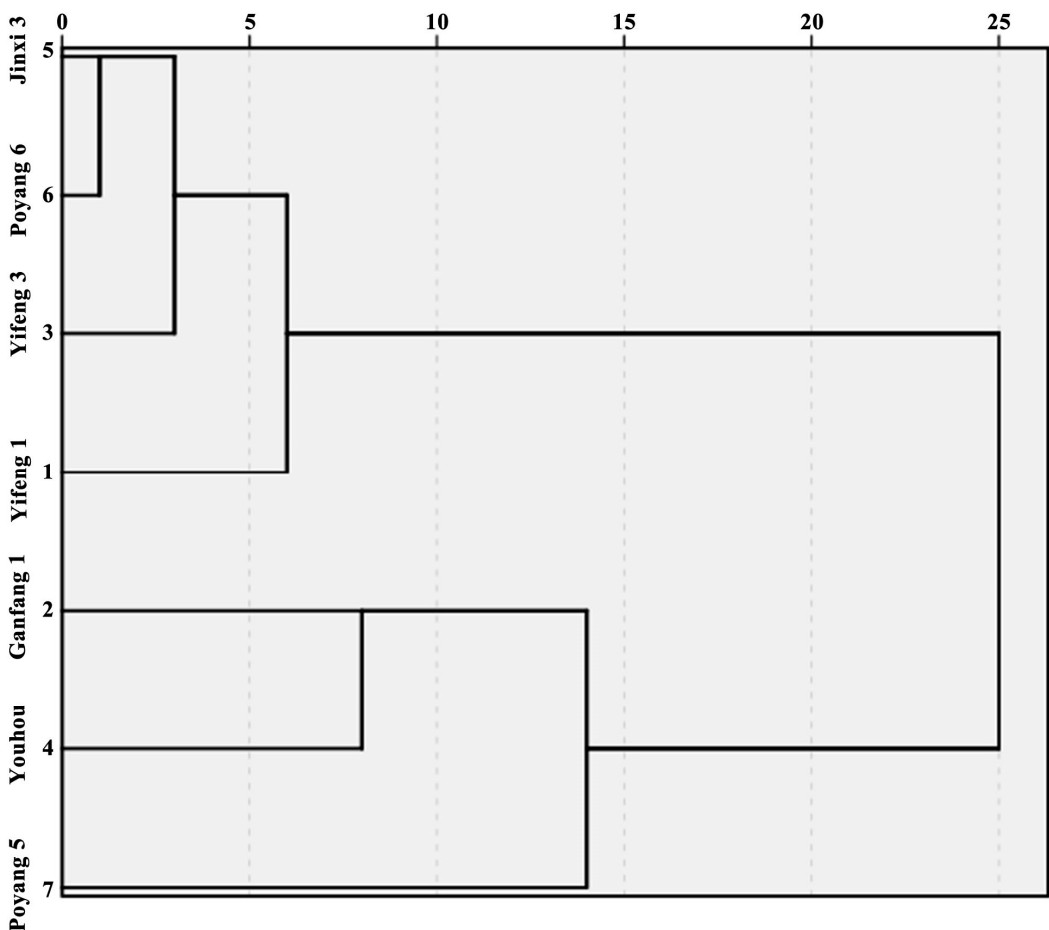

**Figure 7.** Cluster analysis of different persimmon cultivars.

### 3.6. Expression of Adversity and Stress Genes in Different Persimmon Varieties

*CBF* transcription factors play critical roles in resistance to cold stress [23]. To evaluate the cold resistance of the seven different persimmon varieties, qRT—PCR analysis of *DkCBF* was carried out, as shown in Figure 8A. The expression level of *DkCBF* in 'Jinxi 3' was significantly higher than that of the other varieties, which indicated that the cold resistance was strongest in 'Jinxi 3' and that the expression level of *DkCBF* in 'Ganfang 1', 'Poyang 5', 'Yifeng 1', and 'Yifeng 3' was similar to or higher than that of 'Poyang 6' and 'Youhou'. This meant that the cold resistance of 'Ganfang 1', 'Poyang 5', 'Yifeng 1', and 'Yifeng 3' was higher than that of 'Poyang 6' and 'Youhou'.

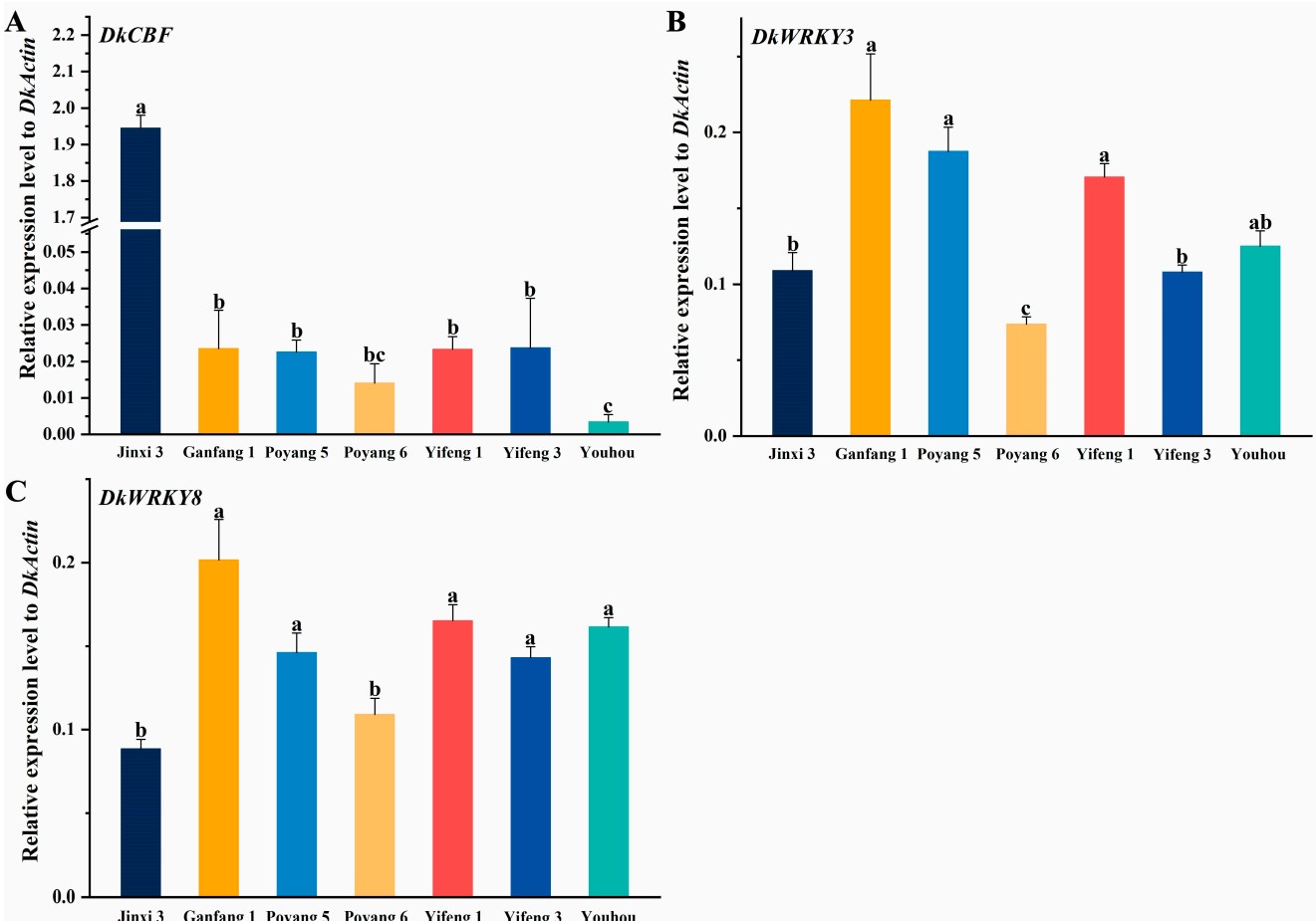

**Figure 8.** Expression of the genes involved in cold resistance (**A**) and anthracnose resistance in different persimmon varieties (**B**,**C**). Means with different letters indicate significant differences at $p < 0.05$ by Tukey's test.

*DkWRKY3/8* are effective for the plant to resist anthracnose [24]. The expression levels of *DkWRKY3/8* were all highest in 'Ganfang 1'. The expression level of *DkWRKY3* in 'Poyang 5' was lowest, and the lowest expression level of *DkWRKY8* presented in 'Jinxi 3' (Figure 8B,C). This result suggested that 'Ganfang 1' may have the strongest resistance to anthracnose.

### 3.7. Expression of PA Pathway Genes in Different Persimmon Varieties

The PA pathway genes *DkANR*, *DkLAR*, and *DkLAC* were evaluated with qRT–PCR [25]. As shown in Figure 9, the expression level of *DkANR* in 'Ganfang 1' was moderate, and that of *DkLAR* and *DkLAC* was more modest. This result is consistent with the fact that the PA content in 'Ganfang 1' was moderate among the seven varieties.

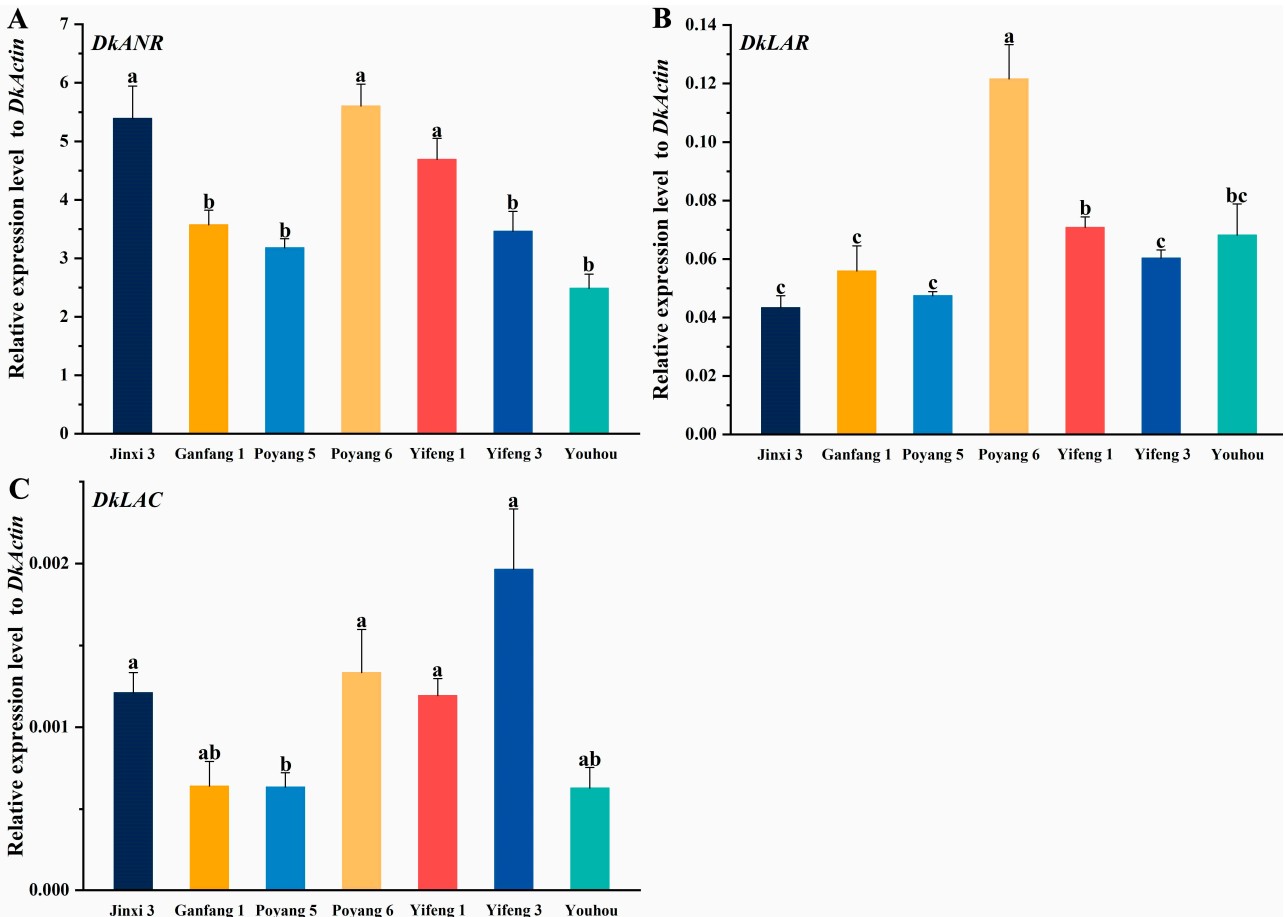

**Figure 9.** Expression of the genes involved in the PA pathway in different persimmon varieties. (**A**). *DkANR*; (**B**). *DkLAR*; (**C**). *DkLAC*. Means with different letters indicate significant differences at $p < 0.05$ by Tukey's test.

The 'Youhou' fruit is oblate and shows a high and stable yield, few seeds, good quality, stress tolerance, and parthenocarpy. It is suitable to be stored and transported, and the soluble solids content in the fruit is generally over 18%. It has recently become increasingly popular in the market. However, the genetic improvement of PCNA, whose parents are both J-PCNA persimmons, entails the dilemma of obvious inbreeding depression [26,27]. As the natural deastringency trait in J-PCNA is controlled by a recessive gene [28,29], the conventional breeding period is longer (12–19 years). Thus, it is highly necessary to expand the range of breeding parents for breeding novel PCNA persimmon, which is important for the genetic improvement of persimmon. In the Chinese PCNA (C-PCNA), the natural deastringency trait is controlled by a dominant gene [30,31], and the C-PCNA has a relatively distant relationship to the J-PCNA [32] and is a more ideal parent for PCNA breeding. Unfortunately, the number of C-PCNA genotypes is limited; in particular, there are fewer male germplasm types. Therefore, it is hoped that the non-PCNA persimmon can be used as the female parent and the C-PCNA persimmon as the male parent for cross-breeding. For example, research has shown that a new PCNA genotype can be obtained through the cross between the androecious genotype Male 8 and a non-PCNA female [33]. This demonstrated that a PCNA progeny with the natural deastringency trait of C-PCNA could be obtained by using the non-PCNA type as the parent. This has an important meaning and wide prospects for the genetic improvement of C-PCNA.

Our study compared and comprehensively evaluated fruit quality by measuring the fruit exterior quality, nutrient content, and functional ingredient content, providing a more scientific reference for the selection of superior varieties. 'Ganfang 1', also called 'Gaoan

wuhefangshi', is a seedless single plant selected from the asexual progeny of 'Gaoan fangshi' that has large, square, seedless, single fruit with a weight reaching 138–180 g. It is the ideal material to serve as the main extension variety and breeding parent. Developing persimmon fruit with fine qualities has been a goal in several breeding programs, but postharvest traits have rarely been examined. In this study, we evaluated commercially important postharvest traits in seven different persimmon cultivars. As shown in Table 2, the fruit weight of 'Ganfang 1' is just lighter than that of 'Yifeng 1', reaching 177.350 g FW, the fruit shape index is 2.567, which is less than those of 'Jinxi 3' and 'Poyang 6', and the fruit shape is rounder. The vitamin C content is 209.524 mg/100 g FW, which is higher than that of 'Poyang 6', 'Yifeng 3', and 'Yifeng 1'; the soluble sugar concentration is 0.038 FW% lower than that of 'Yifeng 3', 'Yifeng 1', 'Youhou', and 'Poyang 5'. The titratable acid content is 1.973 FW% higher than that of 'Youhou', 'Yifeng 1', 'Yifeng 3', and 'Jinxi 3', so it is possible to know that the sugar-acid ratio of 'Ganfang 1' is lower than that of 'Yifeng 1', 'Yifeng 3', and 'Youhou'. Moreover, the flavor of 'Ganfang 1' fruit is better. As a comprehensive evaluation result, 'Ganfang 1' fruit quality is close to that of the 'Youhou' fruit.

In addition, stress resistance is an important index for evaluating the comprehensive quality of persimmon varieties. Freezing injury and disease are common field stresses on persimmon. The C-repeat binding factor (*CBF*) gene family controls cold tolerance in plants, and its members are well conserved among eudicots and monocots, among which there are diverse homologs [34]. In *Arabidopsis thaliana*, *CBFs* are reported to play a dominant role in the cold-responsive network by directly regulating the expression levels of cold-responsive (*COR*) genes [35]. To analyze the cold resistance of the seven persimmon varieties, we screened *DkCBF* (C-repeat binding factors) for qRT—PCR analysis. After analyzing the *CBF* gene expression of the different persimmon varieties, 'Jinxi 3' was found to have the strongest cold resistance, and the cold resistance of the other persimmon varieties was weaker. Anthracnose is a fungal destructive disease that seriously damages persimmon fruits, leaves, and branches. Screening for persimmon varieties resistant to anthracnose is one of the goals of persimmon variety breeding. WRKY transcription factors are reported to be involved in resistance to anthracnose. We isolated two WRKY transcription factors (*WRKY3/8*) for qRT—PCR analysis and compared the resistance to anthracnose of seven persimmon varieties. It was found that the expression of *DkWRKY3/8* was the highest in 'Ganfang 1' (Figure 8), indicating that 'Ganfang 1' had the strongest resistance to anthracnose. Moreover, we also detected the expression level of *DkWRKY5/7* in the seven persimmon varieties, and the results showed that it was not significantly expressed in 'Ganfang 1' (Supplementary Materials Figure S2), which indicated that *DkWRKY5/*7 may not contribute to regulating the resistance to anthracnose in 'Ganfang 1.'

PAs are high-molecular-weight polyphenol polymers that specifically accumulate in the vacuoles of tannin cells in persimmon pulp and are important components influencing the quality of persimmon fruit. Moreover, the *ANR*, *LAR*, and *LAC* genes are members of the PA-specific pathway for PA biosynthesis. By comparing the expression levels of these genes, the tannin synthesis ability of different persimmon varieties can be evaluated. As a result, the PA biosynthesis abilities of 'Ganfang 1' and 'Poyang 5' were deemed weak in the seven persimmon varieties. In addition, the *DkDFR* and *DkANS* genes, which belong to the core flavonoid-anthocyanin pathway, were analyzed by qRT—PCR, revealing a moderate expression level in 'Ganfang 1' among the seven persimmon varieties (Supplementary Materials Figure S3).

Our study aimed to screen for an excellent persimmon parental germplasm in Jiangxi Province. The results of this study showed that 'Ganfang 1' owed a good advantage due to hybrid parents, which resulted in an attractive appearance and potential for strong adaptability. In addition, the non-PCNA fruit is especially suitable for making dried persimmon, so 'Ganfang 1' may have highly important application potential in the processing of persimmon byproducts.

## 4. Conclusions

In our study, the measurement of 13 quality indicators for seven different persimmon varieties was completed within 48 h, and the expression of the genes involved in resistance to biotic and abiotic stress and in PA metabolism was analyzed. Outliers were removed from the quality indicator data after they were standardized. The external quality (fruit weight, horizontal length, vertical length, and firmness) and internal quality (titratable acid content, vitamin C content, flavonoid content, anthocyanin content, soluble sugar content, pectinase activity, soluble protein content, tannin content, and tannin cell size) indicators of the seven persimmon germplasms were calculated with variant analysis, principal component analysis, and cluster analysis. When compared to J-PCNA 'Youhou', we found that 'Poyang 5' and 'Ganfang 1' exhibited a strong comprehensive quality and rated just below 'Youhou'. Furthermore, 'Ganfang 1' was resistant to biotic and abiotic environmental stress, particularly from anthracnose. Taken together, our findings suggest that 'Ganfang 1' has significant promotional and breeding potential for persimmon improvement.

**Supplementary Materials:** The following supporting information can be downloaded at: https://www.mdpi.com/article/10.3390/horticulturae8090844/s1, Table S1: The persimmon germplasms distribution in Jiangxi Province; Table S2: Analysis of persimmon orchard soil physical and chemical properties in Nanchang City, Jiangxi Province; Table S3: The primers used in this study; Figure S1: Annual average climatic conditions change in Nanchang City, Jiangxi Province (2020); Figure S2: Expression of the *DkWRKY5* and *DkWRKY7* genes in different persimmon varieties; Figure S3: Expression of the *DkDFR* and *DkANS* genes involved in PAs pathway in different persimmon varieties.

**Author Contributions:** Investigation: formal analysis, writing—original draft, methodology, S.Y.; formal analysis, writing—original draft, M.Z. (Meng Zhang); investigation, M.Z. (Ming Zeng) and M.W.; conceptualization, resources, Z.L. and X.H.; formal analysis, supervision, writing—review and editing, Q.Z. All authors have read and agreed to the published version of the manuscript.

**Funding:** This research was supported by the National Key R&D Program of China (2019YFD1000600) and Jiangxi Academy of Agricultural Sciences Ph.D. Start-up Fund (2121241).

**Institutional Review Board Statement:** Not applicable.

**Informed Consent Statement:** Not applicable.

**Data Availability Statement:** Data are contained within the article.

**Acknowledgments:** The authors thank Pingxian Zhang (from Huazhong Agricultural University, China) and Changfei Guan (from Northwest A&F University, China) for their suggestions and improvement of this manuscript.

**Conflicts of Interest:** The authors declare no conflict of interest.

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
