# Peer review of "Assessment of Fruit Quality and Genes Related to Proanthocyanidins Biosynthesis and Stress Resistance in Persimmon (Diospyros kaki Thunb.)"

_horticulturae, doi:10.3390/horticulturae8090844_

Round 1

Reviewer 1 Report

The manuscript titled „Assessment of Fruit Quality and Genes Related to Proanthocya- nidins and Stress Resistance in Persimmon (Diospyros kaki Thunb.)” is about a quality indicators’ is an evaluation in the persimmon population derived from China and Japan. The observed 13 fruit quality indicators represent well the fruit characteristics of persimmon. Growing of persimmon is increasing not just in Asia, but worldwide year on year at present. I think this manuscript is a great study, it contains a lot of useful data, which can be used in the near future. Is it possible to have any detailed data about the material? E.g. about thier phenology? It would be better to add a short description about them. Why have you selected the observed cultivars? What was the reason to select them? I think if you check the fruit data, it is neccessary to add the climate and soil conditions, from where the samples derived. Please add these data to be able to evaluate the observed results.

Author Response

Response to Reviewer 1 Comments

  • The manuscript titled “Assessment of Fruit Quality and Genes Related to Proanthocyanidins and Stress Resistance in Persimmon (Diospyros kaki)” is about a quality indicators’ is an evaluation in the persimmon population derived from China and Japan. The observed 13 fruit quality indicators represent well the fruit characteristics of persimmon. Growing of persimmon is increasing not just in Asia, but worldwide year on year at present. I think this manuscript is a great study, it contains a lot of useful data, which can be used in the near future. Is it possible to have any detailed data about the material? E.g. about thier phenology? It would be better to add a short description about them.

Response: Thanks for your suggestion. We had added a short description about the material’s phenology in the L98-101 as “The budding period is around February 24 every year, and the flowering period is around April 26 every year. The tree is open and the growth potential is strong. Except for ‘Yifeng 1’ and ‘Yifeng 3’, which are monoecious, both bear only female flowers.”

  • Why have you selected the observed cultivars? What was the reason to select them? I think if you check the fruit data, it is neccessary to add the climate and soil conditions, from where the samples derived. Please add these data to be able to evaluate the observed results.

Response: Thanks for your kind suggestion. The reason we have selected the observed cultivars is that they are the main and special varieties of astringent persimmons in Jiangxi Province, with a large cultivation area and production. Moreover, they have attractive appearance quality, ‘Youhou’ is currently one of the most popular and commercial Japanese sweet persimmon variety with the most comprehensive traits, so ‘Youhou’ is adopted as a reference variety. We have added the description about the climate and soil conditions, from where the samples derived, also we revised the sentence in L91-92 as “the climate and soil conditions from where the samples derived were showed in the Figure S3 and Table S3, respectively”.

Thanks very much for your kind comments.

Reviewer 2 Report

Refer the stage of fruit collection

RNAisoPlus please refer to the company and not the code of the product

qPCR please provide the primers used for each gene analysed, for the quantitative analysis table S1 is not provided please provide, also describe the method for the quantitative analysis is it ΔΔCt?

The authors state that CBF transcription factor is key gene involved in the resistance to cold stress and this is the reason for studying this gene, however the authors did not perform an experiment for cold stress so why the differences in the expression in the different varieties correlates to cold stress tolerance and not to any other function ?

Author Response

Response to Reviewer 2 Comments

(1)RNAisoPlus please refer to the company and not the code of the product

Response: Thanks for your suggestion. We revised the description in L145-146 as “The total RNA was isolated from fruit flesh using RNAiso Plus* (Tiangen, Beijing, China)”.

(2) qPCR please provide the primers used for each gene analysed, for the quantitative analysis table S1 is not provided please provide, also describe the method for the quantitative analysis is it ΔΔCt?

Response: Thanks for your question. We have provided all the primers used for each gene analyzed in the Table S2. Yes, we used the ΔΔCt method for quantitative analysis, as follows:

  1. The ct value of the target gene minus the ct value of the internal reference gene.
  2. The ct value of the test sample minus the ct value of the control sample.
  3. Negative value of the difference takes the logarithm.

 (3) The authors state that CBF transcription factor is key gene involved in the resistance to cold stress and this is the reason for studying this gene, however the authors did not perform an experiment for cold stress so why the differences in the expression in the different varieties correlates to cold stress tolerance and not to any other function?

Response: Thanks for your question. The CBF can function as a transcriptional activator that binds to the C-repeaty/DRE DNA regulatory element and, thus, is likely to have a role in cold- regulated gene expression in Arabidopsis (Stockinger et al., 1997). So the CBF could be thought as a transcriptional activator that involved in the resistance to cold stress. In this study, the samples were subjected to a cold stress treatment on dry ice before being treated with liquid nitrogen. So via the analysis of the expression level of CBF, we could predict the resistance of different persimmon varieties to cold stress.

Stockinger, E.J., Gilmour, S.J., Thomashow, M.F. Arabidopsis thaliana CBF1 encodes an AP2 domain-containing transcriptional activator that binds to the C-repeaty/DRE, a cis-acting DNA regulatory element that stimulates transcription in response to low temperature and water deficit. Proc. Natl. Acad. Sci. 1997, 94, 1035-1040.

(4) The methods described must be improved.

Response: Thanks for your suggestion. We have added the detail description of methods in the L114-L116 as “The extracting solution was titrated with 0.1 mol/L NaOH standard solution after phenolphthalein reagent was added.”, in the L120-122 as “The extract was added with 1 ml 9 % phenol solution and 5 ml concentrated sulfuric acid successively. After standing at room temperature for 30 min, the color was meas-ured at 485 nm.”, in the L124-125 as “Take 1 ml extract, add 5 ml Coomassie Brilliant Blue G-250, mix and stand for 2 min, measure absorbance at 595 nm.”, in the L127-129 as “The flesh was extracted with 1% hydrochloric acid-methanol solution and the absorb-ance was measured at 325 nm and 530 nm. The contents of flavonoids and anthocya-nins were calculated respectively.”, in the L132-137 as “The pulp was grinded into homogenate and mixed with 2 ml of 0.4 % pectin solution. The mixture was reacted in a 45 °C water bath for 30 min. Then 1.5 ml of 3,5-Dinitrosalicylic acid (DNS) reagent was added and heated in a boiling water bath for 5 min. After cooling, the solution was diluted to 25 ml. The absorbance was meas-ured at 520 nm. The amount of D-galacturonic acid was determined based on the standard curve.”

Thanks very much for your kind comments.
